# Altered salience network structure–function integration underlies the decline in cognitive flexibility during aging

Xing Qian[1], Wan Lin Yue[1,2], Kwun Kei Ng[1], Ruth L. F. Leong[1], Fang Ji[1], Narayanaswamy Venketasubramanian[3], Saima Hilal[4], Christopher Chen[4], Michael W. L. Chee[1], Dani S. Bassett[5], Juan Helen Zhou [1,2,6,7]*

**1** Centre for Sleep and Cognition & Centre for Translational Magnetic Resonance Research, Yong Loo Lin School of Medicine, National University of Singapore, Singapore, Singapore, **2** Integrative Sciences and Engineering Programme, NUS Graduate School, National University of Singapore, Singapore, Singapore, **3** Raffles Neuroscience Centre, Raffles Hospital, Singapore, Singapore, **4** Memory Aging and Cognition Centre, Department of Pharmacology, Yong Loo Lin School of Medicine, National University of Singapore, Singapore, Singapore, **5** Departments of Physics & Astronomy, Bioengineering, Electrical & Systems Engineering, Neurology, and Psychiatry, University of Pennsylvania, Philadelphia, Pennsylvania, United States of America, **6** Department of Electrical and Computer Engineering, National University of Singapore, Singapore, Singapore, **7** Department of Medicine, Healthy Longevity & Human Potential TRP, Yong Loo Lin School of Medicine, National University of Singapore, Singapore, Singapore

* helen.zhou@nus.edu.sg

## Abstract

Cognitive flexibility supports efficient switching between mental sets and contributes to the preservation of general cognition in aging. It relies on the integration between brain functional dynamics and structural architecture. However, how this structure–function integration changes with age and contributes to cognitive flexibility decline in older adults remains unclear. In this study, we investigated longitudinal aging-related changes in multimodal structure–function integration, quantified as functional signal alignment (i.e., coupling) versus liberality (i.e., decoupling) relative to individual structural connectomes, which represent distinct spectral components, and tested their longitudinal associations with cognitive flexibility. Resting-state fMRI signals were decomposed based on diffusion MRI–derived structural networks using a graph signal processing framework. We focused on subnetworks within three core large-scale cognitive systems: the executive control network (ECN), default mode network (DMN), and salience network (SN). Across two independent datasets, the task-positive SN-A subnetwork, which includes core SN regions such as the anterior insula and dorsal anterior cingulate cortex, exhibited decreased coupling and increased decoupling with aging. Importantly, these changes were associated with a greater decline in cognitive flexibility (measured by the Trail Making Test and Color Trails Test) over time. In contrast, task-negative DMN-A (centered in the medial prefrontal and posterior cingulate cortex) showed aging-related changes in the opposite direction, with increased coupling and decreased decoupling over time. Together, these findings reveal network-specific trajectories of intrinsic structure–function integration

**Data availability statement:** The processed numerical data underlying the figures can be found within the paper and/or Supporting information files. The raw data supporting the findings of this study are not publicly available due to ethical and privacy restrictions associated with human participant data. Data access requests related to the Singapore Longitudinal Brain Aging Study (primary dataset) can be submitted to the Centre for Sleep and Cognition, National University of Singapore (https://medicine.nus.edu.sg/csc/contact-us/). Data access requests related to the Harmonization Study (validation dataset) can be submitted to the Memory, Ageing and Cognition Centre, National University of Singapore (https://medicine.nus.edu.sg/macc-2/projects/harmonization-study/). Access to these data will be considered upon reasonable request and subject to approval by the relevant institutional and ethics committees. Custom analysis code used in this study is available at https://github.com/hzlab/2026_Qian_PLOS_BIOL_Coupling_Aging (DOI https://doi.org/10.5281/zenodo.19316674).

**Funding:** This study was supported by the National Medical Research Council, Singapore (NMRC/OFLCG19May-0035 to JHZ; NMRC/CIRG/1485/2018 to JHZ; NMRC/CSA-SI/0007/2016 to JHZ; NMRC/MOH-00707-01 to JHZ; NMRC/CG/435 M009/2017-NUH/NUHS to JHZ; CIRG21nov-0007 to JHZ; HLCA23Feb-0004 to JHZ; OFYIRG23jul-0010 to XQ), the RIE2020 AME Programmatic Fund from A*STAR, Singapore (A20G8b0102 to JHZ), the Ministry of Education, Singapore (MOE-T2EP40120-0007 to JHZ; T2EP2-0223-0025 to JHZ; MOE-T2EP20220-0001 to JHZ), and the Yong Loo Lin School of Medicine Research Core Funding, National University of Singapore (to JHZ). The funders had no role in study design, data collection and analysis, decision to publish, or preparation of the manuscript.

**Competing interests:** The authors have declared that no competing interests exist.

**Abbreviations:** CSF, cerebrospinal fluid; CTT, Color Trails Test; DMN, default mode network; dMRI, diffusion-weighted MRI; ECN, executive control network; FC, functional connectivity; GFT, graph Fourier transform; GSR, global signal regression; MPRAGE, magnetization prepared rapid acquisition gradient echo; ROI, regions of interest; SC, structural connectivity; SN, salience network; TMT, Trail Making Test.

in normal aging and indicate that preserved structure–function integration within the SN may be particularly important for maintaining cognitive flexibility in older adults.

## Introduction

Cognitive flexibility is a core executive function that enables efficient switching between different perspectives, goals, or tasks, thereby supporting adaptive thinking and behavior in response to changing circumstances [1–3]. It relies on the coordinated operation of multiple executive processes, including salience detection, attention, working memory, inhibition, and cognitive switching [4,5]. In older adults, preserved cognitive flexibility is particularly important for maintaining independence and quality of life, whereas declines in flexibility can contribute to broader age-related cognitive difficulties and reduced daily functioning [6–9]. Understanding the neural mechanisms that support cognitive flexibility is therefore essential for developing effective strategies to mitigate cognitive decline and promote healthy aging.

Normal aging is accompanied by widespread alterations in brain structure and function, including degeneration of white matter pathways [10–12], reductions in grey matter integrity [13–19], and changes in functional brain activity [20–25]. At the network level, older adults typically show reduced within-network specialization and diminished segregation between functional networks, particularly within the executive control network (ECN), default mode network (DMN), and salience network (SN) [26–30]. These networks are critically involved in higher-order cognitive processes that are known to decline with age [26,29,31,32]. Specifically, the ECN supports cognitive control functions and goal-directed behavior [32,33], the DMN is implicated in internally directed cognition, including self-referential thought and autobiographical memory [34], and the SN plays a central role in detecting salient stimuli and orchestrating dynamic switching between the ECN and DMN [33,35]. This latter function enables the SN to mediate the shift between externally oriented attention and internally focused mental states, a mechanism that becomes increasingly important for cognitive flexibility in aging.

In parallel, diffusion MRI studies consistently report reduced structural connectivity (SC) and network efficiency with aging, with decreases both within and between network modules [36–39], and converging evidence suggests that age-related functional connectivity (FC) and SC alterations show partially overlapping spatial patterns and are related to one another [26,27,40,41]. Importantly, age-related disruptions in both functional and SC within and between the core cognitive networks are associated with poorer cognitive performance, suggesting that coordinated structure–function alterations may contribute to cognitive decline in aging [11,14,26,29,42–44]. Because neuroanatomical connectivity constrains functional interactions, the relationship between SC and FC has been widely studied as an index of structure–function coupling [45–53]. It has been widely examined as a marker of coordinated brain organization and its disruption in aging and disease [27,41,54–59].

However, the simple correlation between SC and FC offers a mechanistic, yet incomplete, explanation of brain function [48,60,61]. Recent graph signal processing

approaches instead quantify structure–function integration by directly characterizing how functional signals are constrained by, or deviate from, the underlying structural connectome. Using a graph Fourier framework, BOLD activity can be decomposed into structurally aligned versus structurally liberal components, which represent distinct spectral components [62,63]. Specifically, alignment reflects stronger structural constraints on intrinsic functional dynamics (i.e., higher coupling), whereas liberality reflects greater deviations from structural architecture (i.e., higher decoupling). Importantly, these indices differ from conventional SC–FC coupling measures because they are derived from graph spectral decomposition of BOLD signals with respect to the structural connectome, rather than from correlations between SC and FC. To improve accessibility, we refer to alignment and liberality as coupling and decoupling, respectively, while noting that these terms are defined within the present graph-spectral framework and are distinct from correlation-based coupling/decoupling metrics. Notably, in young adults performing a cognitive switching task, better performance was associated with reduced signal liberality/decoupling, suggesting that tighter structural constraints may support cognitive flexibility [62]. This framework therefore offers a promising avenue for testing whether age-related alterations in intrinsic structure–function integration contribute to longitudinal decline in cognitive flexibility [64–68], which has been linked to both structural and functional brain changes in older adults [69–71]. In parallel, prior work has demonstrated that cognitive flexibility across the life span is supported by large-scale brain dynamics [72]. However, how the functional dynamics constrained by the underlying structural connectome change in aging remains largely unexplored.

Accordingly, the present study examined longitudinal changes in intrinsic structure–function integration (coupling and decoupling) at the network level in normal aging and tested whether these changes predict longitudinal decline in cognitive flexibility [measured by out-of-scanner Trail Making Test (TMT) and Color Trails Test (CTT)] in older adults. Since most prior structure–function integration studies in aging are cross-sectional, making it difficult to distinguish true within-person change from cohort effects, our longitudinal approach allows us to quantify within-individual trajectories of structure–function coupling/decoupling. Particularly, we assessed intrinsic structure–function integration from resting-state fMRI, providing a task-independent characterization of aging-related changes that complements prior task-based work. We focused on the subnetworks of ECN, DMN, and SN because they form core large-scale systems supporting higher-order cognitive function including cognitive flexibility and executive control, show robust age-related alterations in functional segregation and SC, and the SN in particular coordinates switching between ECN and DMN [26–30]. We hypothesized that functional signals would become progressively less coupled (or more decoupled) with respect to the underlying neuroanatomical architecture over time, and that greater deviations from normative integration would be associated with steeper cognitive flexibility decline.

## Methods

### Participants

Two independent datasets comprising healthy older adults were used to investigate aging-related changes in structure–function integration. The primary dataset was drawn from the neuroimaging subset of the Singapore-Longitudinal Aging Brain Study (S-LABS) [73]. Participants were included only if they had at least two time points of neuroimaging data that passed quality control procedures (see Image processing), with inter-scan intervals ranging from 18 to 24 months over a 5-year period. To ensure a healthy aging cohort, participants were excluded if they met any of the following criteria at any time point: (1) Mini-Mental State Examination score <26; (2) modified-Geriatric Depression Screening Scale score ≥9; (3) history of significant vascular events; (4) history of malignant neoplasia; (5) history of organ failure (heart, lung, liver, or kidney); (6) thyroid disease (active or inadequately treated); (7) existing neurological or psychiatric conditions; or (8) history of head trauma accompanied by loss of consciousness. These exclusion criteria were consistent with previous studies using the same dataset [29,42,73]. The final sample consisted of 54 healthy older adults (24 females; age range: 59–82 years; mean baseline age = 66.7 ± 4.9 years). Of these, 35 participants had data from two time points, while 19 participants had data from three time points.

A second dataset of healthy older adults, described in a previously published study [74,75], was used for independent replication and validation of the primary findings. Participants were excluded based on the following criteria: (1) history of stroke; (2) MRI evidence of cerebrovascular disease (defined by MRI markers: presence of cortical infarcts and/or ≥ 2 lacunes and/or confluent white matter lesions in two brain regions with Age Related White Matter Changes score ≥8); (3) diagnosis of cognitive impairment from review of clinical measures and neuropsychological assessments; and (4) less than two time points of neuroimaging data that passed quality control criteria. The final validation sample included 39 healthy older adults (29 females; age range: 62–80 years; mean baseline age = 68.9 ± 5.1 years). Of these, 27 participants had data from two time points, and 12 participants had data from three time points (mean interval = 2.7 ± 1.0 years).

Ethical approval for the two studies was obtained from the Institutional Review Board of the National University of Singapore (H-17-077) and the Domain Specific Review Board of the National Healthcare Group, Singapore (2010/00017), respectively. All participants provided written informed consent prior to participation. The study was conducted in accordance with the ethical principles of the Declaration of Helsinki.

### Neuropsychological assessments

To assess cognitive flexibility in the main dataset, a subset of 52 participants (24 females, mean baseline age = 66.7 ± 4.9 years) completed Parts A and B of the TMT [14,73,76,77] within three months of each neuroimaging session. All participants had at least two time points with both neuroimaging and TMT data (mean interval = 2.6 ± 1.0 years). In brief, TMT A required participants to sequentially connect the numbers 1–25, which were randomly distributed across a sheet of paper. TMT B introduced a cognitive switching component by requiring participants to alternate between numbers (1–13) and letters (A–L) in ascending order (e.g., 1–A–2–B…). According to the concept of global switching cost, defined as the increase in response time for switching versus non-switching tasks [66,67], we calculated dTMT, the difference in completion time between TMT B and TMT A (dTMT = TMT B − TMT A), as a measure of cognitive flexibility. These dTMT values were standardized into T scores by first z-scoring and then transforming them to a distribution with a mean of 50 and a standard deviation of 10, for use in statistical analyses.

Cognitive flexibility in the validation dataset was assessed using Parts A and B of the CTT [78–80], which was developed as an alternative to TMT. CTT A, which is structurally identical to TMT A, required participants to sequentially connect numbers 1–25. CTT B involved the same numerical sequence but added a switching demand by presenting the numbers in two different colors, requiring participants to alternate between colors while maintaining numeric order. As with the main dataset, the difference in completion time between CTT B and CTT A (dCTT = CTT B − CTT A) was used to index cognitive flexibility. These dCTT values were also converted to T scores and used in the statistical analyses.

### Image acquisition

Participants from both the main and validation datasets were scanned using a 3T Siemens Tim Trio system. For the main dataset, high-resolution T1-weighted structural scans were acquired using a magnetization prepared rapid acquisition gradient echo (MPRAGE) sequence with the following imaging parameters: 192 continuous sagittal slices, TR = 2,300 ms, TE = 2.98 ms, TI = 900 ms, flip angle = 9°, FOV = 256 × 256 mm², voxel size = 1.0 × 1.0 × 1.0 mm³. High-resolution T2-weighted structural scans were also acquired with the following imaging parameters: 192 continuous sagittal slices, TR = 3,200 ms, TE = 448.0 ms, TI = 1,800 ms, flip angle = 0°, FOV = 256 × 256 mm², voxel size = 1.0 × 1.0 × 1.0 mm³. Resting state fMRI scans (8 min) were acquired for the main dataset with participants fixating on a cross presented in the center of the screen (36 continuous axial slices, TR = 2,000 ms, TE = 30 ms, flip angle = 90°, FOV = 192 × 192 mm², voxel size = 3.0 × 3.0 × 3.0 mm³). Diffusion-weighted MRI (dMRI) scans for the main dataset were acquired using a single-shot spin echo planar imaging sequence with the following imaging parameters: 54 continuous axial slices, TR = 9 600 ms, TE = 107 ms, FOV = 256 × 256 mm², voxel size = 2.0 × 2.0 × 2.0 mm³, 30 non-collinear diffusion gradient directions at $b$ = 1,000 s/mm², 6 volumes of $b$ = 0 s/mm².

In the validation dataset, high-resolution T1-weighted structural scans were acquired using the same MPRAGE sequence as the main dataset (except TE = 1.9s). A shorter resting state fMRI scan (5 min) with the same fixation instruction was collected (48 continuous axial slices, TR = 2,300 ms, TE = 25 ms, flip angle = 90°, FOV = 192 × 192 mm$^2$, voxel size = 3.0 × 3.0 × 3.0 mm$^3$). The dMRI scans were acquired with a different set of imaging parameters (48 continuous axial slices, TR = 6,800 ms, TE = 85 ms, FOV = 256 × 256 mm$^2$, voxel size = 3.0 × 3.0 × 3.0 mm$^3$, 61 non-collinear diffusion gradient directions at $b$ = 1,150 s/mm$^2$, 7 volumes of $b$ = 0 s/mm$^2$). For the validation dataset, fluid attenuated inversion recovery scans were also collected for assessment of cerebrovascular disease markers (48 continuous axial slices, TR = 9,000 ms, TE = 82 ms, flip angle = 180°, FOV = 256 × 256 mm$^2$, voxel size = 1.0 × 1.0 × 3.0 mm$^3$).

## Image processing

Structural and functional images from both the main and validation datasets were preprocessed using standard protocols implemented in FSL [81] and AFNI [82], following procedures established in prior studies [11,29,42]. In brief, preprocessing of T1 images involved reduction of image noise (SUSAN), skull stripping (Brain Extraction Tool), registration to the Montreal Neurological Institute (MNI) 152 standard space (linear and non-linear using FLIRT and FNIRT, respectively), and segmentation of the brain into grey matter, white matter, and cerebrospinal fluid (CSF).

Preprocessing of fMRI images involved removal of the first five volumes, correction for slice timing and motion, skull stripping, spatial smoothing (Gaussian kernel of 6 mm full-width half maximum), temporal band pass filtering (0.009–0.1 Hz), removing first and second order trends, co-registration with structural image (Boundary-Based Registration), non-linear registration to standard space (FNIRT), and nuisance signal regression (CSF, white matter, global signal and 6 motion parameters). Global signal regression (GSR) was applied in the main pipeline to reduce global shared fluctuations. To assess robustness, we repeated the key analyses using the same preprocessing pipeline without GSR.

Preprocessing of dMRI images involved correction of eddy current distortion and head motion through affine registration of dMRI images to the first $b$ = 0 volume. Rotation of diffusion gradients was performed to compensate for motion. An additional distortion correction step using T2 images was performed for the main dataset. Fractional anisotropy (FA) images were obtained by voxel-wise fitting of a diffusion tensor model to the diffusion data (DTIFIT).

Quality control procedures were applied to exclude participants with excessive head motion. For fMRI, participants were excluded if maximum absolute displacement exceeded 4 mm or maximum framewise displacement exceeded 1 mm. For dMRI, participants were excluded if their maximum displacement relative to the first b = 0 volume exceeded 3 mm. Additionally, all fMRI and dMRI images were visually inspected for accurate co-registration with corresponding T1-weighted structural images.

## Structural network construction and BOLD time series extraction

An overview of the methodology used in this study is illustrated in Fig 1. Using a network parcellation of 126 regions of interest (ROIs), consisting of 114 cortical ROIs defined by Yeo and colleagues [83], which can be grouped into seven large-scale networks (DMN, ECN, SN, limbic network, dorsal attention network, somatomotor network and visual network), and 12 subcortical ROIs grouped into a subcortical network [84], BOLD signal time series was extracted for each ROI from the preprocessed fMRI of each individual. Given that the limbic network has been reported to exhibit the lowest test–retest reliability [85,86], its ROIs were excluded from further analysis. To assess the robustness of our findings, we repeated the analyses in the main dataset using an alternative brain parcellation with 416 ROIs, constructed by combining the Schaefer's 400-parcel (17-network) cortical parcellation with the Tian's 16-parcel subcortical atlas [87,88].

To construct the structural network from the preprocessed DTI data for each individual, the voxel-wise probabilistic distribution of fiber direction was first modeled using Bayesian estimation of diffusion parameters (bedpostX) [89]. Next, the predefined 122 ROIs (after removal of ROIs from the limbic network) from the same brain ROI parcellation template were used for probabilistic fiber tracking [90] in the Pipeline for Analyzing braiN Diffusion imAges (PANDA) toolbox [91].

 

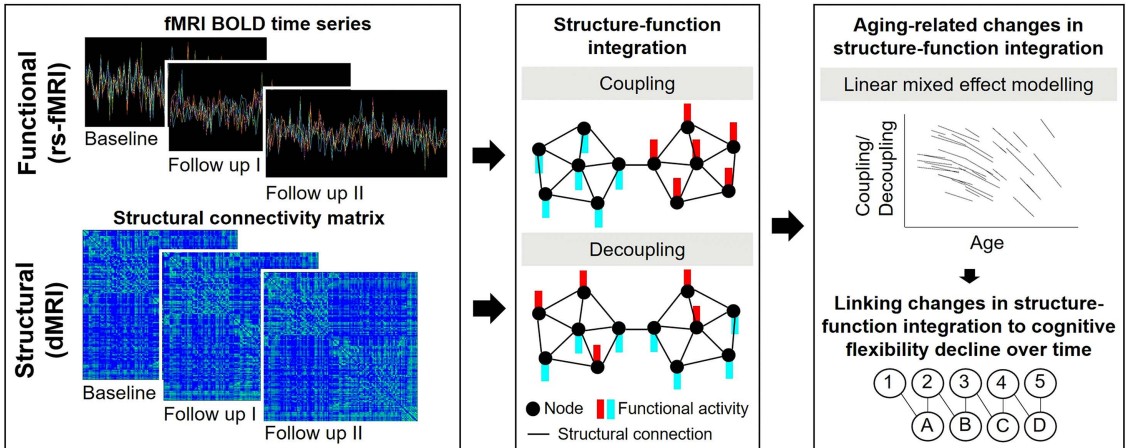

**Fig 1. Overview of the study design.** Neuroimaging data were collected at two to three time points per participant. Resting-state functional MRI (rs-fMRI) was used to extract time series of brain activity, while diffusion-weighted MRI (dMRI) was used to construct structural connectivity (SC) matrices. Using the graph Fourier transform, functional signals were decomposed into a portion that was aligned (coupled) with the underlying structural network (i.e., signals that vary smoothly across densely connected nodes), and another portion that was liberal (decoupled), deviating from the structural network (i.e., signals that vary sharply between densely connected nodes). In the illustration, SC between nodes (i.e., regions of interest [ROIs]) is shown as black edges, and the amplitude of BOLD fMRI signals at each ROI is represented by colored bars. Longitudinal changes in coupling and decoupling were modeled using linear mixed-effects models and evaluated in relation to cognitive flexibility decline over time.

One ROI with insufficient brain coverage (<95%) in the DTI images was dropped across participants (also removed from fMRI for consistency), resulting in a 121 ROI × 121 ROI SC matrix for each participant. Since the estimated probabilities between any two ROIs are different depending on the ROI from which the fiber tracks initiated, we averaged the two unidirectional connectivity probabilities as connection probability $P_{ij}$ for each pair of ROIs [36]. To account for differences in overall connection density and magnitude across participants (i.e., network cost), and to facilitate inter-subject comparability, the raw connection probabilities $P_{ij}$ between the $i$th ROI and $j$th ROI in the symmetrical matrix were log-transformed and normalized [56,92]. The final normalized weight $w_{ij}$ was calculated as:

$$w_{ij} = \frac{\log(P_{ij}) - \min[\log(P_{ij})]_{1<i\neq j<n}}{\max[\log(P_{ij})]_{1<i\neq j<n} - \min[\log(P_{ij})]_{1<i\neq j<n}},$$

(1)

where $n$ is the number of ROIs. The resulting individual structural network served as the anatomical network upon which the resting-state fMRI signals from the same participant were projected and quantified for degree of coupling and decoupling.

## Derivation of structure–function coupling and decoupling

Measures of structure–function coupling and decoupling were derived following the graph signal processing framework proposed by Medaglia and colleagues [62]. For clarity and accessibility, we use the terms coupling and decoupling to refer to the alignment and liberality indices in the original study. Importantly, these measures should not be conflated with conventional structure–function coupling metrics based on SC–FC correlations, as they are computed via graph spectral decomposition of BOLD signals with respect to the structural connectome.

Specifically, eigenvector decomposition was performed on the symmetric structural network $A \in R^{n \times n}$, where $n$ is the number of nodes (i.e., ROIs), to obtain a set of eigenvalues and corresponding eigenvectors:

$$A = V\Lambda V^T,$$

(2)

where $\Lambda$ denotes the diagonal matrix of eigenvalues, ordered as $\lambda_0 \leq \lambda_1 \leq \ldots \leq \lambda_{n-1}$, and $V = \{v_k\}_{k=0}^{n-1}$ is the matrix of associated eigenvectors. These eigenvectors form the spatial frequency basis of the graph $A$: "spatially rough" components (corresponding to negative eigenvalues) reflect highly connected nodes possess values of different signs (i.e., signal variation is abrupt across the network), whereas "spatially smooth" components indicate strongly connected nodes possess values of same signs (i.e., signal variation is coherent across the network).

A graph Fourier transform (GFT) was then applied to the fMRI BOLD time series $x \in R^n$, defined over brain ROIs, projecting the signals onto the structural eigenbasis:

$$\widetilde{x} = V^T x = [\widetilde{x}_0, \; \widetilde{x}_1, \ldots, \widetilde{x}_{n-1}]^T.$$

(3)

The original signal can be reconstructed as:

$$x = V\widetilde{x} = \sum_{k=0}^{n-1} \widetilde{x}_k v_k.$$

(4)

This transformation yields a spatial frequency-domain representation of the fMRI signal, where each GFT coefficient $\widetilde{x}_k$ quantifies the contribution of a specific eigen vector $v_k$. The weighted "spatially smooth" components reflect the portion of functional signals aligned with the underlying structural network, whereas the weighted "spatially rough" components indicate the portion of functional signals deviating liberally from the structural network.

To extract alignment (coupling) and liberality (decoupling) measures which represent distinct spectral components, we applied graph filters to retain only the most structurally aligned (coupled) or liberal (decoupled) components. $K_A$ and $K_L$ denote the number of eigenvectors retained for the coupled and decoupled components, respectively. Rather than selecting a fixed number of graph spectral components or applying a spectrum dichotomization threshold as in prior studies [62,63], we determined the coupling and decoupling components using a variance-based criterion derived from the structural eigenvalue spectrum, which provides a principled way to define low- and high-frequency subspaces and is robust across parcellations with different ROI counts. In brief, for each group of the "spatially rough" and "spatially smooth" components, corresponding to negative and positive eigenvalues, respectively, we computed the total spectral energy by summing the absolute values of eigenvalues. The eigenvalues were then sorted in descending order within each group, and the cumulative sum was calculated. The value of $k_A$ (or $k_L$) was defined as the minimum number of top eigenvalues required to explain 95% of the total spectral energy within that group. This procedure was repeated for each time point, and the resulting $k_A$ (or $k_L$) values were averaged across time points within each subject to yield subject-specific values for $K_A$ and $K_L$. This approach allowed us to ensure that the selected eigenvectors represented the most informative smooth and rough components of the structural graph spectrum, while maintaining consistency across individuals and time points.

For implementation, we used the 95% variance threshold as the main analysis and repeated the procedure using 80% and 90% thresholds to assess robustness. For the main dataset, the average numbers of retained eigenvectors were: $K_A$= 7, 12, 16 and $K_L$ = 54, 66, 75; for the validation dataset, $K_A$ = 8, 13, 18 and $K_L$= 52, 65, 73 (corresponding to 80%, 90%, and 95% variance explained, respectively).

For each participant, the coupled and decoupled signal components were averaged across time and ROIs to produce static network-level measures of structure–function coupling and decoupling. In addition to averaging the coupled and decoupled signal components across time points to obtain static, network-level measures, we conducted a sensitivity analysis using an alternative summary metric commonly adopted in prior studies [63,93], namely the L2-norm of the component vectors. Network-level coupling and decoupling were re-computed in the main dataset (126-ROI parcellation)

using the L2-norm to quantify component magnitude. Because the L2-norm is sign-invariant, we applied a constant shift to the coupled/decoupled component time series to ensure non-negativity prior to computing the norm.

We focused on the ECN, DMN, and SN—three large-scale brain systems consistently implicated in age-related functional decline [30–34]. In the 17-network parcellation defined by Yeo and colleagues [83], each of these large-scale systems—ECN, DMN, and SN—is subdivided into finer-grained subnetworks that reflect spatially and functionally distinct components. The ECN comprises three subnetworks: ECN-A, primarily involving dorsolateral prefrontal and inferior parietal regions; ECN-B, which includes lateral prefrontal and posterior middle temporal areas; and ECN-C, a more medial and frontal subnetwork that is close to default-mode regions. The DMN includes DMN-A, centered in medial prefrontal and posterior cingulate cortex; DMN-B, which extends into lateral temporal cortex; and DMN-C, a ventromedial prefrontal subsystem. The SN is subdivided into SN-A and SN-B. SN-A primarily involves the anterior insula and dorsal anterior cingulate cortex, while SN-B encompasses regions such as the frontal operculum, temporal-parietal junction, and adjacent lateral prefrontal cortex, and is more closely associated with stimulus-driven attentional reorienting, environmental monitoring, and bottom-up attention capture. To control for multiple comparisons, we restricted the analysis to 3 subnetworks in ECN, 3 in DMN, and 2 in SN, with coupling and decoupling computed for each, resulting in 16 total comparisons.

As a control analysis to contextualize the specificity of effects observed in higher-order cognitive networks, we extended the longitudinal analyses (95% variance) to include the somatomotor and visual networks in addition to the ECN, DMN, and SN subnetworks.

## Estimation of aging-related changes in structure–function integration in older adults

To investigate the longitudinal changes in structure–function coupling and decoupling, we modeled the effects of time on network-level structure–function coupling and decoupling in the main and validation datasets separately using linear mixed effect models, following our previous work [29,42]:

$$Y_{ij} = \beta_{00} + \beta_{01} (\text{Gender}_j) + \beta_{02} (\text{Education}_j) + \beta_{03} (\text{Age}_j) + \beta_{10} (\text{Time}_{ij}) + \beta_{11} (\text{Age}_j * \text{Time}_{ij}) + \mu_{0j} + \mu_{1j} (\text{Time}_{ij}) + r_{ij}, \quad (5)$$

where $Y$ represented structure–function coupling (or decoupling) from a given network for each participant at each visit; Gender was a binary variable; Age and Education were grand-mean centered baseline age and years of education; Time represented time interval since the first (baseline) session; $\beta$ terms represented fixed effects and $\mu$ terms represented random effects for each participant; Subscripts $i$ and $j$ represented visit and participant, respectively. The fixed-effect coefficient for Time was interpreted as an estimate of the longitudinal (within-subject) effect on structure–function coupling or decoupling. Bonferroni correction was applied across 16 comparisons (8 subnetworks × 2 metrics). In addition, we also applied FDR correction across subnetwork-level tests to provide a complementary, less conservative control of multiple comparisons.

To provide an intuitive summary of the magnitude of longitudinal effects, we additionally quantified the variance explained by the time-related fixed effects using a marginal $R^2$ ($R^2m$) framework for mixed-effects models [94]. Specifically, we fit a reduced model by removing the time and baseline age × time fixed-effect terms from Equation (5), while keeping the same random-effects structure. We then computed the incremental explained variance as $\Delta R^2m = R^2m(\text{full}) - R^2m(\text{reduced})$, which represents the additional variance explained by the time-related fixed effects. In parallel, we performed likelihood ratio tests comparing the full and reduced models. To facilitate interpretation, $\Delta R^2m$ values were used as effect-size summaries of time-related longitudinal change and were reported alongside $\beta$ coefficients.

To provide additional insights, we also conducted an exploratory ROI-level analysis of structure–function coupling/decoupling within the ECN, DMN, and SN subnetworks. For each ROI, longitudinal effects were tested using linear mixed-effects models consistent with the main analyses, and ROI-level p-values were corrected for multiple comparisons using Bonferroni correction across ROIs within these networks.

To provide background for interpreting structure–function integration results, we additionally computed mean intrinsic BOLD activity within each ECN, DMN, and SN subnetwork and tested longitudinal effects using the same linear mixed-effects models and related the longitudinal changes in intrinsic subnetwork BOLD activity to corresponding changes in cognitive flexibility.

**Association of structure–function integration with cognitive flexibility**

We next examined if longitudinal changes in structure–function coupling and decoupling were associated with corresponding changes in cognitive flexibility. We restricted our analysis to subnetworks that showed significant time effects in the previous longitudinal models. Following previous work [29], we estimated individual longitudinal change rates ($B_{1j}$) of both structure–function coupling (or decoupling) and cognitive flexibility for each participant using a simple regression model:

$$Y'_{ij} = B_{0j} + B_{1j}\left(\text{Time}_{ij}\right) + r_{ij},\tag{6}$$

where $Y'_{ij}$ represents the predicted values of structure–function coupling, decoupling, or cognitive flexibility (dTMT scores in the main dataset and dCTT scores in the validation dataset), derived from the linear mixed-effects model described in Equation (5). The slope term $B_{1j}$ reflects each participant's rate of change over time.

We then tested whether longitudinal changes in structure–function coupling or decoupling were associated with longitudinal change in cognitive flexibility. This effect was modeled as:

$$B_{1j.dTMT} = b_0 + b_1\left(B_{1j.\text{Coupling}}\right),\tag{7}$$

$$B_{1j.dTMT} = b_0 + b_1\left(B_{1j.\text{Decoupling}}\right).\tag{8}$$

All statistical analyses were carried out in MATLAB 2015a (The MathWorks).

## Results

### Network-level structure–function integration showed differential longitudinal changes in older adults

In the main dataset, longitudinal changes in structure–function integration were observed within the SN, ECN, and DMN, particularly in the SN-A, DMN-A, and ECN-C subnetworks (Fig 2 and Table A in S1 Appendix). Consistent with our hypothesis, SN-A exhibited a significant decrease in coupling ($p = 0.027$) and a concurrent increase in decoupling ($p = 0.008$) over time, suggesting that functional signals became progressively less constrained by the underlying structural network with aging.

In contrast, DMN-A and ECN-C displayed an opposite pattern: coupling increased ($p < 0.001$ for DMN-A, $p = 0.018$ for ECN-C), while decoupling decreased ($p < 0.001$ for DMN-A, $p = 0.009$ for ECN-C) over time. Notably, ECN-C is anatomically proximal to regions typically associated with the DMN (Fig 2B), making it unsurprising that ECN-C exhibited similar directional changes as DMN-A. These findings may reflect a divergent trajectory of aging-related changes in structure–function integration between task-negative networks (e.g., DMN-A and ECN-C) and task-positive networks (e.g., SN-A).

Of these effects, only DMN-A survived Bonferroni correction for multiple comparisons across coupling and decoupling metrics in all subnetworks (adjusted $\alpha = 0.05/16 \approx 0.003$). Nonetheless, all observed trends were robust across different values of $K$ used in the graph filtering procedure (Table A in S1 Appendix). In addition to Bonferroni-corrected $p$-values, uncorrected and FDR-adjusted $p$-values are reported for subnetwork analyses (see Tables A and B in S1 Appendix).

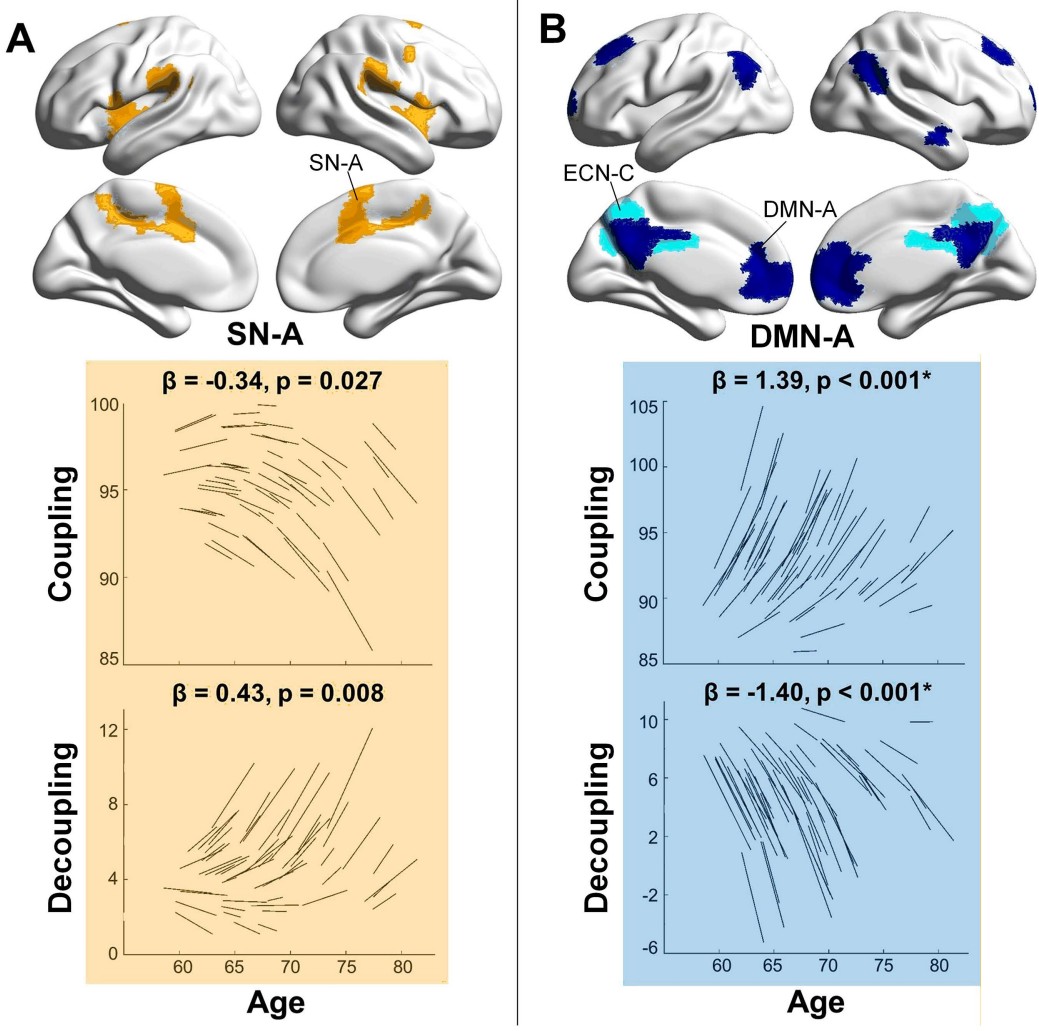

**Fig 2. Differential changes in structure–function coupling and decoupling with aging in the main dataset.** Longitudinal effects were estimated using linear mixed-effects models, with fixed coefficients ($\beta$) representing the effect of time interval from the first scan. Spaghetti plots display individual trajectories of structure–function coupling and decoupling over time, adjusted for baseline age, gender, and education. Coupling and decoupling measures were computed using *K* values explaining 95% of variance, shown here for illustration; similar results were observed with 80% and 90% variance thresholds. Asterisks (*) indicate effects that survived Bonferroni correction for multiple comparisons ($a = 0.05/16 \approx 0.003$). Brain network visualizations were created using BrainNet Viewer (http://www.nitrc.org/projects/bnv). The underlying numerical data for this figure are provided in Supporting information (S1 Data).

For structure–function coupling, the additional variance explained by the time-related fixed effects ($\Delta R^2 m$) was 0.025 for SN-A, 0.127 for DMN-A, and 0.032 for ECN-C. Likelihood ratio tests comparing the full and reduced models yielded p-values of 0.025, <0.001, and 0.061, respectively. For structure–function decoupling, the corresponding $\Delta R^2 m$ values were 0.052 for SN-A, 0.163 for DMN-A, and 0.049 for ECN-C, with model comparison *p*-values of 0.022, <0.001, and 0.033, respectively.

These findings were partially replicated in the validation dataset. Specifically, SN-A showed a significant decline in coupling over time ($p = 0.016$), and along with a trend toward increased decoupling ($p = 0.072$). Additionally, DMN-B exhibited a significant increase in coupling over time ($p = 0.026$) (Fig 3; Table B in S1 Appendix).

PLOS Biology

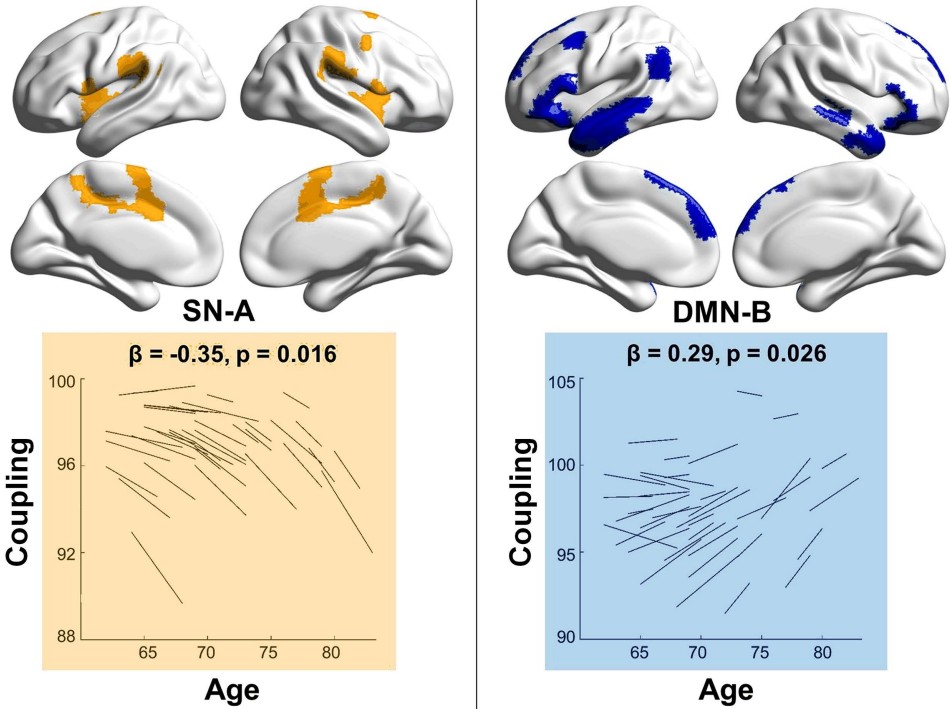

**Fig 3. Differential changes in structure–function coupling and decoupling with aging in the validation dataset.** Longitudinal effects were estimated using linear mixed-effects models, with fixed coefficients ($\beta$) representing the effect of time interval from the first scan. Spaghetti plots display individual trajectories of structure–function coupling and decoupling over time, adjusted for baseline age, gender, and education. Coupling and decoupling measures were computed using $K$ values explaining 95% of the observed variance, shown here for illustration; similar results were observed with 80% and 90% variance thresholds. Asterisks (*) indicate effects that remained statistically significant after Bonferroni correction for multiple comparisons ($\alpha = 0.05/16 \approx 0.003$). Brain network visualizations were created using BrainNet Viewer (http://www.nitrc.org/projects/bnv). The underlying numerical data for this figure are provided in Supporting information (S2 Data).

Exploratory ROI-level analyses in main dataset revealed patterns broadly consistent with the network-level findings, with the strongest aging-related coupling/decoupling changes observed in ROIs within the SN-A and DMN-A subnetworks (e.g., SalVentAttnA_FrMed and DefaultA_PFCm; Fig C in S1 Appendix).

In a control analysis extending the subnetwork-level longitudinal models in main dataset to include somatomotor and visual networks (in addition to the ECN, DMN, and SN subnetworks), we found that after applying FDR and Bonferroni corrections, the sensorimotor systems did not exhibit significant longitudinal changes in structure–function coupling or decoupling, whereas SN-A, DMN-A, and ECN-C remained significant after correction, indicating robust longitudinal effects. These control analyses provide additional context supporting the relative specificity of the strongest coupling/decoupling alterations in higher-order cognitive and salience/attention-related systems (Table E in S1 Appendix).

In a control analysis examining longitudinal changes in mean SC and intrinsic BOLD activity within ECN, DMN, and SN subnetworks (Table F in S1 Appendix), only intrinsic BOLD activity in DMN-B and SN-A showed significant longitudinal change, with opposite directions (DMN-B increased, whereas SN-A decreased). When relating longitudinal changes in intrinsic BOLD activity in DMN-B and SN-A to corresponding changes in cognitive flexibility, no significant associations were observed ($p = 0.058$ and $p = 0.18$, respectively). Compared with these modest component-level changes, structure–function integration indices showed stronger longitudinal effects, suggesting higher sensitivity to aging-related change in this dataset.

Sensitivity analyses without GSR yielded longitudinal time effects that were virtually identical to the primary results (Table D in S1 Appendix), suggesting that our conclusions are robust to the use of GSR.

**Longitudinal changes in cognitive flexibility were associated with structure–function coupling and decoupling changes in the salience network**

Longitudinal changes in cognitive flexibility were significantly associated with longitudinal alterations in structure–function coupling and decoupling within SN-A in both datasets (Fig 4). In the main dataset, a slower rate of decline in cognitive flexibility (as measured by dTMT) was associated with a smaller decrease in SN-A coupling and a smaller increase in SN-A decoupling over time ($b = 1.69$, $p < 0.001$ and $b = -0.84$, $p = 0.002$, respectively). These findings are consistent with

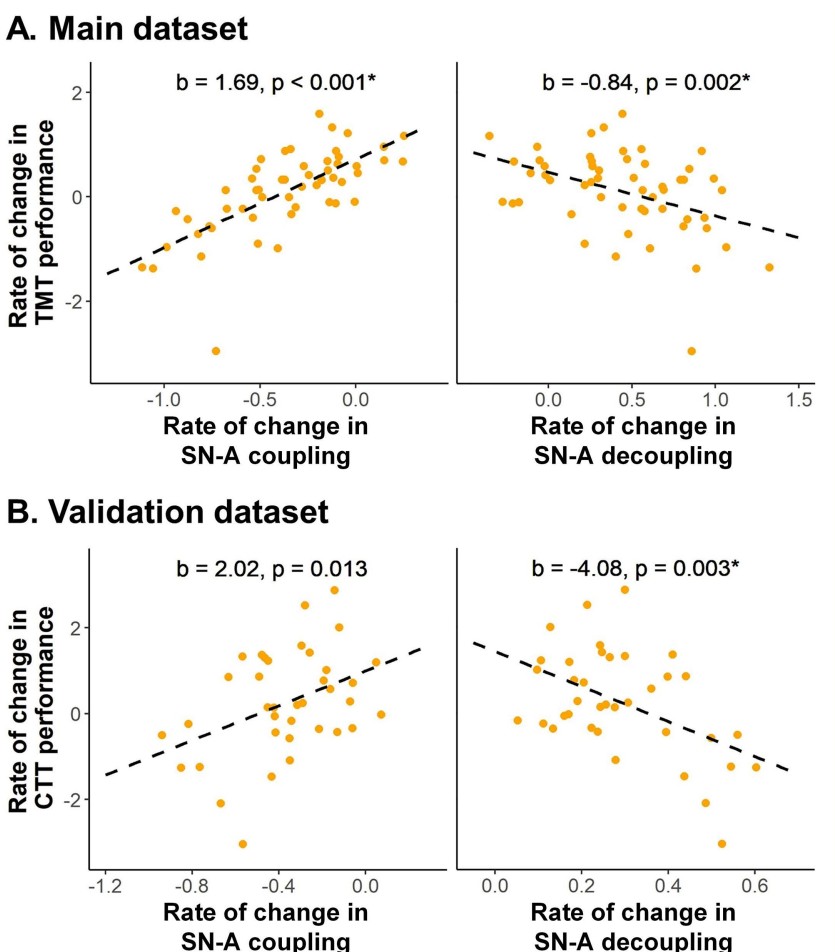

**Fig 4. Cognitive flexibility decline over time is associated with changes in salience network structure–function coupling and decoupling in matin and validation datasets.** Associations between longitudinal change rates in cognitive flexibility, measured by the Trail Making Test (TMT) in the main dataset (panel **A**) and the Color Trails Test (CTT) in the validation dataset (panel **B**), and change rates in structure–function integration are shown. Scatter plots illustrate the relationship between re-estimated individual slopes ($\beta_{1j}$) in TMT (or CTT) performance and coupling/decoupling metrics, adjusted for baseline age, gender, and education. Coupling and decoupling measures were derived using $K$ values explaining 95% of the variance; similar results were observed with 80% and 90% thresholds. Asterisks (*) indicate associations that remained statistically significant after Bonferroni correction for multiple comparisons ($\alpha = 0.05/7 \approx 0.007$). The underlying numerical data for this figure are provided in Supporting information (S3 Data).

 PLOS Biology

previous work in young adults showing that reduced decoupling is associated with lower switch cost during cognitive flexibility tasks [62].

These associations were robust across different values of $K$, and they remained statistically significant after multiple comparisons correction (Table C in S1 Appendix). Importantly, these findings were replicated in the validation dataset, where a slower decline in cognitive flexibility (as indexed by dCTT scores) was associated with a slower decrease in SN-A coupling and a slower increase in SN-A decoupling ($b = 2.02$, $p = 0.013$ and $b = -4.08$, $p = 0.003$, respectively).

Sensitivity analyses using the L2-norm summarization approach yielded longitudinal effects and brain–behavior associations for structure–function coupling that were largely consistent with the main results (Fig B in S1 Appendix).

When replicating the analyses in the main dataset using an alternative parcellation comprising 416 ROIs, the overall findings remained largely consistent with the primary results (Fig A in S1 Appendix). Specifically, the SN-A subnetwork exhibited decreased structure–function coupling and increased decoupling over time, and these longitudinal changes were significantly associated with cognitive flexibility decline as measured by the TMT. In contrast, the DMN-A subnetwork showed the opposite pattern, with increased coupling and decreased decoupling over time. However, ECN-C did not show significant longitudinal effects under the alternative parcellation ($\beta \approx 0$, $p > 0.80$). This discrepancy may reflect reduced robustness of the ECN-C effect across parcellations, potentially due to differences in how this relatively small subnetwork is represented (e.g., altered spatial boundaries), which may increase measurement noise and reduce sensitivity to detect subtle longitudinal changes.

## Discussion

Age-related alterations in FC and SC have been widely reported, including reduced SC integrity/efficiency and reduced functional segregation across large-scale systems. Our graph-spectral coupling/decoupling metrics are not intended to replicate these established findings; rather, they provide complementary information by quantifying how the structural connectome constrains intrinsic functional dynamics. In this way, coupling/decoupling indices may offer an additional and potentially more sensitive perspective on aging-related reorganization beyond conventional FC or SC measures alone.

Using two independent longitudinal datasets, we investigated aging-related changes in intrinsic structure–function integration within the subnetworks of three core large-scale cognitive networks, the ECN, DMN, and SN, and examined how these changes relate to longitudinal decline in cognitive flexibility. Overall, we found that both structure–function coupling and decoupling changed over time in normal aging. Notably, these changes followed opposing trajectories in task-positive versus task-negative networks: SN-A exhibited decreasing coupling and increasing decoupling, whereas DMN-A, subnetwork with strong associations to task-negative processing, showed increasing coupling and decreasing decoupling. Crucially, a smaller decline in SN-A coupling and a smaller increase in SN-A decoupling were associated with a slower decline in cognitive flexibility, suggesting that preservation of normative structure–function integration patterns within the SN-A may be critical for maintaining cognitive flexibility in older adults. These findings provide novel insights into how aging alters the degree to which functional brain activity is constrained by SC and demonstrate that such changes are meaningfully linked to aging-related cognitive decline.

### Biological interpretation of structure–function coupling and decoupling in aging

Within the graph-spectral framework, structure–function coupling (alignment) and decoupling (liberality) can be interpreted in relation to low-dimensional bases of brain organization supported by anatomical connectivity. Structural eigenmodes provide spatial bases constrained by the white-matter connectome, such that low-frequency (smooth) modes represent spatially extended, anatomically supported patterns of activity, whereas high-frequency (rough) modes capture more spatially complex patterns that deviate from structural constraints. Increased coupling therefore reflects a shift of intrinsic functional dynamics toward structurally supported low-frequency modes, while reduced decoupling indicates a diminished

contribution from higher-frequency, structurally liberal modes, suggesting reduced spatial complexity and flexibility of intrinsic dynamics.

This interpretation is consistent with recent work demonstrating that structural degeneration is associated with reduced amplitudes of intrinsic functional gradients and altered inter-gradient relationships, a pattern interpreted as a form of low-dimensional "network collapse" characterized by increasing dominance of anatomically constrained activity patterns [95]. In this context, the aging-related increase in coupling and decrease in decoupling observed in DMN-A—robust across the main analysis, sensitivity analyses using a finer parcellation resolution, and ROI-level modeling—may reflect a shift toward structurally dominated intrinsic dynamics in higher-order associative systems, which rely on distributed integration and flexible reconfiguration to support executive function.

## Maintenance of normative salience network structure–function integration in aging supports cognitive flexibility

Among the observed aging-related changes in structure–function coupling and decoupling, the longitudinal time effects in SN-A were consistent with our hypothesis, showing a decline in coupling and an increase in decoupling with age. While prior studies have reported mixed findings regarding age-related functional changes in the SN [96–100], there is more consistent evidence of gray matter loss and white matter tract deterioration in SN regions with aging [98,101–103]. This structural degradation may force greater reliance on indirect anatomical pathways for signal transmission [27,97], potentially reflecting a compensatory shift in network dynamics aimed at preserving SN functionality. Such a shift would manifest as decreased coupling and increased decoupling between functional activity and the underlying structural architecture.

Although structural atrophy and white matter decline are common across multiple brain systems in aging, the SN may be particularly sensitive to disruptions in structure–function integration due to its pivotal role in network switching and brain state transitions [104–106]. Notably, the SN is composed of distinct subnetworks with different functional profiles. SN-A primarily includes regions such as the anterior insula and dorsal anterior cingulate cortex, which are involved in salience detection, interoceptive awareness, and initiating dynamic switching between the DMN and ECN, even during a task-free resting state [107,108]. In contrast, SN-B includes areas such as the frontal operculum and temporoparietal junction, and is more closely associated with stimulus-driven attentional reorienting, bottom-up attention capture, and environmental monitoring. This functional dissociation allows the broader SN to flexibly coordinate internal and external attention based on changing task demands. Our finding of significant aging-related changes in structure–function integration in SN-A, but not SN-B, may therefore reflect differential vulnerability of these SN subsystems to aging-related structural deterioration [31]. While ROI-level findings provide finer-grained localization, we interpret them as exploratory given the multiple-comparison burden and potential parcel-level noise; nevertheless, their consistency with network-level results supports the robustness of our main conclusions.

Importantly, individuals with greater reductions in SN-A coupling and greater increases in decoupling also exhibited steeper declines in cognitive flexibility. These associations align with prior work [62], which found that reduced global liberality (decoupling) was linked to better cognitive switching performance in younger adults. Taken together, these findings support the idea that there is an optimal level of structure–function integration, and that aberrant integration, particularly in the SN, may impair cognitive flexibility. Given the SN's critical role in task-set reconfiguration and network switching, optimal structure–function integration within this network is likely essential for efficiently adapting to task demands. This possibility is further supported by studies showing that SN connectivity correlates with individual differences in cognitive flexibility [109,110], while loss of SN connectivity impairs such abilities [111,112].

## Differential structure–function integration changes in task-positive and task-negative networks in aging

In contrast, longitudinal changes in ECN-C and DMN-A were opposite in direction to those observed in SN-A. Both subnetworks exhibited increased coupling and decreased decoupling with aging. As ECN-C includes regions anatomically

adjacent to the precuneus and posterior cingulate cortex, key nodes of the DMN [34,113], it is plausible that ECN-C shows convergence toward DMN-like organization. Notably, however, this ECN-C effect was smaller in magnitude and was attenuated in sensitivity analyses using a finer parcellation resolution and in ROI-level modeling, suggesting it may be less robust than the DMN-A and SN-A findings. Prior work has suggested that the long-range functional connections of the DMN are disproportionately impacted by aging, leading to increased reliance on short-range connectivity [114,115]. Given that diffusion MRI and tractography may be limited in estimation of longer-range neuroanatomical projections [116], short-range structural connections may contribute more strongly to the SC matrices used in our analysis. Thus, increased coupling and reduced decoupling in the DMN may reflect a greater functional dependency on short-range, structurally mediated connections in aging.

However, the DMN is known to include hub regions that are highly integrated with other networks throughout the brain [117,118]. As such, a shift toward short-range connectivity may signify a less efficient utilization of neural resources, undermining the DMN's ability to coordinate distributed activity and perform complex internal tasks [119,120]. This interpretation is consistent with prior work [27], suggesting that hub-like regions are more susceptible to aging-related declines in communication efficiency. This inefficiency may be further magnified during resting-state conditions, where the DMN is highly active and contributes to maintaining optimal transitions between brain states [121]. As such, the higher coupling and lower decoupling observed in aging DMN networks may underlie reduced functional specialization, with the DMN potentially relying on compensatory recruitment of other networks to maintain its functions [27,29,97]. While we speculated that this pattern may involve greater reliance on short-range connectivity and reduced network efficiency, we emphasize that this interpretation is indirect. Rather, our findings suggest that aging may bias intrinsic functional dynamics in DMN toward more locally coherent, structurally constrained modes, consistent with reduced flexibility of large-scale network organization. Future work integrating graph-spectral measures with explicit metrics of connection length, efficiency, and functional gradients will be important to directly test these mechanistic hypotheses.

Our finding that task-positive SN-A showed the opposite trajectory from DMN-A supports the notion that task-positive and task-negative networks exhibit distinct structural–functional adaptations in aging. We speculate that task-positive networks may require greater functional flexibility to sustain performance in the face of structural decline, thus showing increased decoupling and reduced coupling over time. These compensatory effects may become increasingly important in older adults and manifest as greater divergence between functional signals and structural constraints. By contrast, task-negative networks such as the DMN may shift toward more rigid, structure-driven dynamics, potentially reflecting reduced efficiency in network-level coordination. Additionally, neurovascular differences between task-positive and task-negative networks [122] may further contribute to these divergent trajectories, though future work is needed to explore this possibility in detail.

Our control analyses further indicate that aging-related changes in structure–function integration are network-dependent. Higher-order attention/salience-related systems (e.g., SN and DAN) showed a pattern of reduced coupling and increased decoupling, whereas primary sensorimotor networks showed a different profile (e.g., reduced decoupling with largely stable coupling), consistent with their stronger structural constraints and lower position along the cortical functional hierarchy.

## Limitations and future directions

Several limitations should be acknowledged in the current study. First, although we identified aging-related changes in structure–function integration, most of these effects did not survive correction for multiple comparisons. To mitigate this limitation, we reported only results that were consistent across different $K$ values and partially replicated in an independent dataset, which strengthens the robustness and interpretability of our findings. Second, our sample primarily included older adults, which may limit the generalizability of our findings across the full adult life span. Future studies could incorporate a broader age range to better characterize the trajectories of structure–function integration across the life span and to capture potential critical inflection points in aging-related neural reorganization. Third, vascular health and reactivity may

PLOS Biology

influence the hemodynamic properties of the BOLD signal, particularly in older adults, where neurovascular coupling may be altered [123,124]. Although we attempted to control for this confound by including only participants without a history of major vascular events (main dataset) or cerebrovascular disease (validation dataset), residual vascular effects may still influence estimates of structure–function integration. Future work could incorporate quantitative assessments of vascular health, such as arterial stiffness, cerebral blood flow, or cerebrovascular reactivity, to better isolate neural-specific changes from vascular confounds. Although our study focused on normal aging, the structure–function integration framework may also be valuable for understanding pathological aging and neurodegenerative diseases. Future research could extend this approach by incorporating Alzheimer's disease–related pathologies, such as amyloid burden. In addition, our analyses focused on static, network-level measures of structure–function integration during resting-state fMRI. Future research could extend this work by applying the framework to dynamic or task-based fMRI, particularly in tasks involving cognitive switching. Such investigations could offer a more granular, time-resolved understanding of how anatomical constraints shape neural activity patterns and behavior on a moment-to-moment basis.

## Conclusions

In conclusion, we found that network-level structure–function integration undergoes significant changes with aging, with task-positive SN-A showing decreased coupling and increased decoupling, while task-negative DMN regions exhibited the opposite trend. These divergent trajectories suggest that the degree to which structural networks constrain functional signals in normal aging may be shaped by the functional role of each network, potentially reflecting differences in intrinsic activity, organizational complexity, or efficiency across task-positive and task-negative systems. Importantly, we also observed that age-related changes in SN-A coupling and decoupling were associated with longitudinal changes in cognitive flexibility, underscoring the relevance of structure–function integration for supporting executive function in aging. Together, our findings suggest that aging-related changes in structure–function integration may reflect both network-specific vulnerabilities of the structural connectome and adaptive responses to preserve cognitive performance. These results highlight the value of multimodal analyses for understanding the neurobiological underpinnings of cognitive aging and point to structure–function integration as a promising marker for cognitive resilience in later life.

## Supporting information

**S1 Appendix. Table A.** Age and time effects for structure–function coupling and decoupling of all networks of interest in main dataset. **Table B.** Age and time effects for structure–function coupling and decoupling of all networks of interest in validation dataset. **Table C.** Association between changes in structure–function coupling/decoupling and cognitive flexibility scores in main and validation datasets. **Table D.** Sensitivity analysis without global signal regression (GSR): Age and time effects for structure–function coupling and decoupling in main dataset. **Table E.** Time effects for structure–function coupling and decoupling in main dataset (including sensorimotor networks). **Table F.** Age and time effects for BOLD activity in main dataset. **Fig A.** Longitudinal changes in structure–function coupling and decoupling with aging and their associations with cognitive flexibility decline in the main dataset (validation analysis using a 416-ROI parcellation). **Fig B.** Longitudinal changes in structure–function integration with aging and the associations with cognitive flexibility decline in the main dataset (sensitivity analysis using the L2-norm to quantify component magnitude). **Fig C.** ROI-level longitudinal changes in structure–function integration in the main dataset (exploratory analysis within the DMN, ECN, and SN). (DOCX)

**S1 Data. The underlying numerical data for** Fig 2. (XLSX)

**S2 Data. The underlying numerical data for** Fig 3. (XLSX)

**S3 Data. The underlying numerical data for** Fig 4.
(XLSX)

## Author contributions

**Conceptualization:** Xing Qian, Wan Lin Yue, Michael W.L. Chee, Dani S. Bassett, Juan Helen Zhou.

**Data curation:** Xing Qian, Wan Lin Yue, Kwun Kei Ng, Ruth LF Leong, Fang Ji, Narayanaswamy Venketasubramanian, Saima Hilal, Christopher Chen, Michael W.L. Chee.

**Formal analysis:** Xing Qian, Wan Lin Yue.

**Funding acquisition:** Xing Qian, Juan Helen Zhou.

**Investigation:** Xing Qian.

**Methodology:** Xing Qian, Wan Lin Yue, Kwun Kei Ng, Christopher Chen, Dani S. Bassett, Juan Helen Zhou.

**Project administration:** Xing Qian, Wan Lin Yue, Kwun Kei Ng, Ruth LF Leong, Fang Ji, Narayanaswamy Venketasubramanian, Saima Hilal, Christopher Chen, Michael W.L. Chee.

**Resources:** Juan Helen Zhou.

**Software:** Dani S. Bassett.

**Supervision:** Juan Helen Zhou.

**Validation:** Xing Qian.

**Visualization:** Xing Qian, Wan Lin Yue.

**Writing – original draft:** Xing Qian, Wan Lin Yue.

**Writing – review & editing:** Xing Qian, Wan Lin Yue, Kwun Kei Ng, Ruth LF Leong, Fang Ji, Narayanaswamy Venketasubramanian, Saima Hilal, Christopher Chen, Michael W.L. Chee, Dani S. Bassett, Juan Helen Zhou.

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
