## [Editor Report · Decision Letter 0]

17 Oct 2025

Dear Helen,

Thank you for submitting your manuscript entitled "Altered Salience Network Structure-Function Integration Underlies Cognitive Flexibility Decline in Aging: A Longitudinal Study" for consideration as a Research Article by PLOS Biology. Please accept our apologies for the delay in sending you an initial decision. We had wished to discuss your paper with an Academic Editor with relevant expertise, but had trouble finding someone who was available to provide advice over the last week.

Your manuscript has now been evaluated by the PLOS Biology editorial staff and I am writing to let you know that we would like to send your submission out for external peer review. While we are, in principle, interested in your study, I should add a note that without Academic Editor advice, at this stage we did not feel able to make a firm call about whether the insights provided here offer the level of advance needed for publication at PLOS Biology. We will therefore be looking for strong enthusiasm from the reviewers to move forward with the study.

Please note that before we can send your manuscript to reviewers, we need you to complete your submission by providing the metadata that is required for full assessment. To this end, please login to Editorial Manager where you will find the paper in the 'Submissions Needing Revisions' folder on your homepage. Please click 'Revise Submission' from the Action Links and complete all additional questions in the submission questionnaire.

Once your full submission is complete, your paper will undergo a series of checks in preparation for peer review. After your manuscript has passed the checks it will be sent out for review. To provide the metadata for your submission, please Login to Editorial Manager (https://www.editorialmanager.com/pbiology) within two working days, i.e. by Oct 21 2025 11:59PM.

Kind regards,

Luke

Lucas Smith, Ph.D.

Senior Editor

PLOS Biology

lsmith@plos.org

---

## [Decision Letter · Decision Letter 1]

9 Dec 2025

Dear Helen,

Thank you for your patience while your manuscript "Altered Salience Network Structure-Function Integration Underlies Cognitive Flexibility Decline in Aging: A Longitudinal Study" was peer-reviewed at PLOS Biology, and I apologize again for the protracted review process. Your study has now been evaluated by the PLOS Biology editors, an Academic Editor with relevant expertise, and by several independent reviewers.

In light of the reviews, which you will find at the end of this email, we would like to invite you to revise the work to thoroughly address the reviewers' reports.

As you will see below, the reviewers generally agree that the study offers interesting insights, but the reviewers also provide a number of suggestions to strengthen the study further. We think that the reviewer comments will need to be thoroughly addressed before we can consider your study for publication, and that you will need to go further in justifying the approaches used here, and clarifying how your findings fit with, and extend previous work in this area. We also think that it will be important to conduct additional analyses aimed at improving the robustness of the findings.

Given the extent of revision needed, we cannot make a decision about publication until we have seen the revised manuscript and your response to the reviewers' comments. Your revised manuscript is likely to be sent for further evaluation by all or a subset of the reviewers.

**IMPORTANT - SUBMITTING YOUR REVISION**

*Re-submission Checklist*

*Published Peer Review*

*PLOS Data Policy*

*Blot and Gel Data Policy*

Sincerely,

Luke

Lucas Smith, Ph.D.

Senior Editor

PLOS Biology

lsmith@plos.org

REVIEWS:

Reviewer #1: Cognitive flexibility is a fundamental component of executive function. Cognitive flexibility relies on the integration between brain functional dynamics and structural architecture. This study integrated multimodal MRI data with advanced graph signal processing to address a key question about how the structural architecture constrains functional dynamics in aging. Overall, this work is interesting, but several major and minor concerns remain.

1. The Abstract and Introduction are somewhat redundant and could be more concise. Currently, both sections contain overlapping background information and methodological descriptions, which obscure the study's key message.

2. The authors should clarify the conceptual and methodological distinction between the present work and the previous study (https://www.nature.com/articles/s41562-017-0260-9). Specifically, the manuscript should more clearly articulate how the current longitudinal investigation in aging extends or differs from that prior work in terms of experimental design, analytical framework, and research focus.

3. By applying a graph Fourier transform to decompose resting-state fMRI signals according to the underlying structural connectivity, the authors quantify two complementary indices—alignment and liberality. I would suggest reconsidering the use of the terms "alignment" and "liberality", as they may be somewhat misleading to readers who are not familiar with this specific framework. Using more conventional terminology, such as "coupling" and "decoupling," would improve conceptual clarity and make the findings easier to interpret within the broader context of structure-function integration studies (e.g., https://doi.org/10.1038/s41467-024-46651-8 and https://www.nature.com/articles/s41467-019-12765-7).

4. The authors employed a network parcellation of 126 regions of interest (ROIs), including 114 cortical regions from Yeo et al. and 12 subcortical regions. Although this atlas is well-established, it would be interesting and valuable to assess the robustness of the reported findings using higher-resolution parcellations. For instance, repeating the key analyses with the HCP-MMP1.0+Tian Subcortex atlas or the Schaefer2018_400Parcels_7Networks_order_Tian_Subcortex_S1 parcellation could provide necessary confirmation of the reproducibility of the observed structure-function relationships.

5. The authors derived the alignment and liberality using a variance-based criterion to determine the cut-off (K_A and K_L) in the structural eigenvalue spectrum. This approach differs from prior studies, which typically employed a spectrum dichotomization method (https://www.nature.com/articles/s41467-019-12765-7) or selected only the highest and lowest eigenmodes (https://www.nature.com/articles/s41562-017-0260-9) to represent smooth and rough components. The authors should clarify the conceptual and methodological advantages of the variance-based criterion over these conventional approaches.

6. The authors averaged the aligned and liberal signal components across time points and ROIs to obtain static, network-level measures of structure-function alignment and liberality. In contrast, previous studies have typically used the L2-norm of the component vectors to summarize their magnitude. The authors should clarify the rationale for choosing a simple averaging approach over the L2-norm, and discuss whether this choice could affect the interpretation of alignment and liberality measures (such as https://doi.org/10.1038/s41467-024-46651-8 and https://www.nature.com/articles/s41467-019-12765-7).

7. The authors focused on the executive control network (ECN), default mode network (DMN), and salience network (SN) as primary targets for their analyses. However, the rationale for selecting these three networks over others is not fully articulated. Additionally, exploring age-related structure-function integration at the regional level and linking these regional measures to cognitive performance could provide more detailed insights.

8. The physical meaning of age coefficients (beta) of age-related changes is difficult to comprehend. To provide insight into the overall magnitude and direction of regional age effects, many previous studies have represented the age effects (R-squared values) as the age-related changes during development (see, for example, https://www.nature.com/articles/s41586-022-04554-y and https://www.nature.com/articles/s41593-025-01907-4).

Reviewer #2: This longitudinal study examines brain structure-function integration in aging, focusing on the executive control, default, and salience networks. The authors report differences in alignment between structural and functional network properties of these networks with age and associated them with declines in cognitive flexibility over time. This is an interesting and timely study. Specific comments are below.

In the abstract, the term "liberality" is used before it is defined. Overall, it is not clear why the term "liberality" is introduced at all. This is a confusing term, as it is not typically used in describing functional brain networks. The manuscript and findings would be easier to follow if the authors simply used the term "alignment" (e.g. greater or lesser alignment between functional and structural networks). If increased alignment means decreased liberality, then there is no need for two terms. If they are really measuring different things, this should be more clearly spelled out.

Can the authors comment on the decision to apply global signal regression in the current analyses? Do similar results emerge if GSR is not applied?

The authors may want to reference another relevant study of cognitive flexibility (TMT) and brain dynamics:

Brain Dynamics Underlying Cognitive Flexibility Across the Lifespan. Kupis L, Goodman ZT, Kornfeld S, Hoang S, Romero C, Dirks B, Dehoney J, Chang C, Spreng RN, Nomi JS, Uddin LQ. Cereb Cortex. 2021 Oct 1;31(11):5263-5274.

Reviewer #3: This manuscript presents a sophisticated longitudinal analysis of how the relationship between brain structure and function changes with age and relates to cognitive flexibility. The authors provide evidence that the Salience Network (specifically the SN-A subnetwork) exhibits a distinct pattern of decreasing alignment and increasing liberality with age, and that these changes are associated with a decline in cognitive flexibility. This finding is novel and has significant implications for understanding the neural underpinnings of cognitive aging.

The manuscript is generally well-written, the methods are rigorous, and the conclusions are mostly supported by the data. The study's key strengths include its longitudinal design, the use of two independent datasets for discovery and validation, and the application of a novel graph signal processing technique (graph Fourier transform) to quantify structure-function integration (alignment and liberality). Nonetheless, I have several comments and suggestions that may help further improve the manuscript.

The authors correctly note that only the DMN-A effects survive strict Bonferroni correction across all subnetwork tests. However, for findings that are reproducible across the primary and validation datasets—even if not surviving such strict correction—it may still be meaningful to interpret them cautiously. The authors could also consider applying a less conservative correction method (e.g., False Discovery Rate, FDR) for the primary analysis of the three core networks (ECN, DMN, SN) and their subnetworks. Reporting both uncorrected and FDR-corrected p-values would provide a more nuanced perspective on the results.

Although the authors focus primarily on the three core cognitive networks, it would be valuable to examine age-related changes in structure-function coupling within sensorimotor networks as well. Reporting these results as a control analysis in the Supplementary Information could help contextualize the specificity of the observed effects in higher-order networks.

While the present study focuses on structure-function coupling, it would be informative to first report age-related changes in functional and structural networks separately before presenting the coupling results. Including these analyses in the Supplementary Information would provide a more comprehensive background and help readers interpret the coupling findings.

The observed increase in alignment and decrease in liberality in DMN-A and ECN-C is intriguing but requires further interpretation. The authors suggest this pattern may reflect increased reliance on short-range connections and potentially reduced network efficiency—a plausible explanation, though additional evidence supporting this argument would strengthen the discussion. Furthermore, the concepts of "alignment" and "liberality" are central to the paper but remain somewhat abstract. Although the methodological explanation is clear, a more thorough discussion of the biological interpretation of these metrics would be highly beneficial for the general readership.

In the Introduction, the authors could better emphasize the significance of their study, particularly its longitudinal design, within the existing literature.

---

## [Decision Letter · Decision Letter 2]

10 Mar 2026

Dear Helen,

Thank you for your patience while your manuscript "Altered Salience Network Structure-Function Integration Underlies Cognitive Flexibility Decline in Aging: A Longitudinal Study" was peer-reviewed at PLOS Biology. It has now been evaluated by the PLOS Biology editors, an Academic Editor with relevant expertise, and by two of the original reviewers.

Both reviewers are satisfied by the revision and have recommend that we accept your study (note Reviewer 3 indicated this when submitting his review, but did not provide any specific comments). Based on the reviews, we are likely to accept this manuscript for publication. However, before we can do so we need to you address a number of data and policy-related requests in a last revision that should not take very long. These are detailed below.

**IMPORTANT - Please address the following editorial requests:

1) TITLE: We would like to suggest a tweak to the title of your paper to avoid punctuation, and to streamline it a bit. If you agree, we suggest you shorten your title to:

Altered Salience Network Structure-Function Integration Underlies the decline in Cognitive Flexibility during Aging

2) ETHICS STATMENT: Please update the ethics statement to include the protocol numbers of the protocols approved by the National University of Singapore's IRB. Please also indicate whether the studies were conducted according to the principles expressed in the Declaration of Helsinki.

3) DATA: We understand that you will not be able to share the raw data used here, for legal and privacy reasons. While this is an allowable restriction, under our data sharing policy (see here: http://journals.plos.org/plosbiology/s/data-availability), I do have a few requests to ensure the data is as accessible as possible.

a. Your Data Availability statement currently reads "Data access requests related to the Singapore Longitudinal Brain Aging Study and Harmonization Study can be submitted to the Institutional Data Access and Ethics Committee, NUS Yong Loo Lin School of Medicine, and will be considered upon reasonable request, subject to approval by the relevant institutional and ethics committees." -- Does this apply to both datasets used here (from citations 73 and 74/75)? Please explicitly state this and/or ensure the data availability statement includes the relevant information where each dataset can be accessed.

b. Please update your data availability statement to include more detailed information for who to contact to gain access to the data used in your study. Can you provide websites for the Institutional Data Access and Ethics Committee/ is there a request form somewhere? If not, please provide an email address and as much information as possible about how researchers can request access to those data.

c. If you are legally allowed to do so, we request that you provide the processed data presented in the figures of your study. Note that we do not require all raw data. Rather, we ask that all individual quantitative observations that underlie the data summarized in the figures and results of your paper be made available in one of the following forms:

> Supplementary files (e.g., excel). Please ensure that all data files are uploaded as 'Supporting Information' and are invariably referred to (in the manuscript, figure legends, and the Description field when uploading your files) using the following format verbatim: S1 Data, S2 Data, etc. Multiple panels of a single or even several figures can be included as multiple sheets in one excel file that is saved using exactly the following convention: S1_Data.xlsx (using an underscore).

> Deposition in a publicly available repository. Please also provide the accession code or a reviewer link so that we may view your data before publication.

Regardless of the method selected, please ensure that you provide the individual numerical values that underlie the summary data displayed in your figures.

4) CODE: Per journal policy, if you have generated any custom code during the course of this investigation, please make it available without restrictions. Please ensure that the code is sufficiently well documented and reusable, and that your Data Statement in the Editorial Manager submission system accurately describes where your code can be found. More information on our Code Policy, what and how to share can be found here: https://journals.plos.org/plosbiology/s/code-availability

We expect to receive your revised manuscript within two weeks.

*Published Peer Review History*

*Press*

Sincerely,

Luke

Lucas Smith, Ph.D.

Senior Editor

lsmith@plos.org

PLOS Biology

Reviewer remarks:

Reviewer #1: The authors have addressed all issues.

Reviewer #3, Yong He (editor note: reviewer 3 has signed this review, and recommended 'accept', but did not provide specific reviewer comments):

---

## [Editor Report · Decision Letter 3]

19 Mar 2026

Dear Helen,

Thank you for the submission of your revised manuscript "Altered salience network structure-function integration underlies the decline in cognitive flexibility during aging" for publication in PLOS Biology, and thank you for addressing our last editorial requests in this revision. On behalf of my colleagues and the Academic Editor, Choong-Wan Woo, I am pleased to say that we can in principle accept your manuscript for publication, provided you address any remaining formatting and reporting issues. These will be detailed in an email you should receive within 2-3 business days from our colleagues in the journal operations team; no action is required from you until then. Please note that we will not be able to formally accept your manuscript and schedule it for publication until you have completed any requested changes.

**IMPORTANT - in addition to addressing any production requests, to come, we ask that you address the following editorial requests.

1 - As discussed over email, before we can publish your paper we need you to make your code deposition on GitHub publicly available and to also generate a DOI for that dataset with Zenodo. Once you have completed those steps, please be sure you update your 'data availability statement' in our online system.

2 - I also wanted to let you know that, editorially, we think your paper would be best suited for publication as a 'Short Report' and so I have taken the liberty of changing the article type. Note, this will not require any changes to the manuscript or any updates on your end, but we think the scope of the study is most aligned with that format. You can learn more about the article types here: https://journals.plos.org/plosbiology/s/what-we-publish

If you have any questions about this or would like to discuss, please do get in touch with me.

PRESS

Sincerely,

Luke

Lucas Smith, Ph.D.

Senior Editor

PLOS Biology

lsmith@plos.org